# Anti-*Leptospira* immunoglobulin profiling in mice reveals strain specific IgG and persistent IgM responses associated with virulence and renal colonization

**Frédérique Vernel-Pauillac** [1,2,3], **Gerald L. Murray** [4], **Ben Adler** [4], **Ivo G. Boneca** [1,2,3], **Catherine Werts** [1,2,3] *

**1** Institut Pasteur, Unité Biologie et Génétique de la Paroi Bactérienne, Paris, France, **2** CNRS, UMR 2001, Microbiologie Intégrative et Moléculaire, Paris, France, **3** INSERM, Equipe Avenir, Paris, France, **4** Infection and Immunity Program, Monash Biomedicine Discovery Institute and Department of Microbiology, Monash University, Clayton, Victoria, Australia

* cwerts@pasteur.fr

**Data Availability Statement:** All relevant data are within the manuscript and its Supporting Information files.

## Abstract

*Leptospira interrogans* is a pathogenic spirochete responsible for leptospirosis, a neglected, zoonotic reemerging disease. Humans are sensitive hosts and may develop severe disease. Some animal species, such as rats and mice can become asymptomatic renal carriers. More than 350 leptospiral serovars have been identified, classified on the basis of the antibody response directed against the lipopolysaccharide (LPS). Similarly to whole inactivated bacteria used as human vaccines, this response is believed to confer only short-term, serogroup-specific protection. The immune response of hosts against leptospires has not been thoroughly studied, which complicates the testing of vaccine candidates. In this work, we studied the immunoglobulin (Ig) profiles in mice infected with *L. interrogans* over time to determine whether this humoral response confers long-term protection after homologous challenge six months post-infection. Groups of mice were injected intraperitoneally with $2 \times 10^7$ leptospires of one of three pathogenic serovars (Manilae, Copenhageni or Icterohaemorrhagiae), attenuated mutants or heat-killed bacteria. *Leptospira*-specific immunoglobulin (IgA, IgM, IgG and 4 subclasses) produced in the first weeks up to 6 months post-infection were measured by ELISA. Strikingly, we found sustained high levels of IgM in mice infected with the pathogenic Manilae and Copenhageni strains, both colonizing the kidney. In contrast, the Icterohaemorrhagiae strain did not lead to kidney colonization, even at high dose, and triggered a classical IgM response that peaked at day 8 post-infection and disappeared. The virulent Manilae and Copenhageni serovars elicited high levels and similar profiles of IgG subclasses in contrast to Icterohaemorrhagiae strains that stimulated weaker antibody responses. Inactivated heat-killed Manilae strains elicited very low responses. However, all mice pre-injected with leptospires challenged with high doses of homologous bacteria did not develop acute leptospirosis, and all antibody responses were boosted after challenge. Furthermore, we showed that 2 months post-challenge, mice pre-infected with the attenuated M895 Manilae LPS mutant or heat-killed bacterin were completely protected against renal colonization. In conclusion, we observed a sustained IgM response potentially

**Funding:** This work has been supported by the Laboratoire d'Excellence "Integrative Biology of Emerging Infectious Diseases" Grant ANR-10-LABX-62-IBEID to IGB from the French government's Investissement d'Avenir program. Work at Monash University was supported by grants to BA from the National Health and Medical Research Council (54826) and the Australian Research Council (CE0562063), Canberra, Australia. The funders had no role in study design, data collection and analysis, decision to publish, or preparation of the manuscript.

**Competing interests:** The authors have declared that no competing interests exist.

associated with chronic leptospiral renal infection. We also demonstrated in mice different profiles of protective and cross-reactive antibodies after *L. interrogans* infection, depending on the serovar and virulence of strains.

## Author summary

*Leptospira interrogans* is a pathogenic spirochete responsible for leptospirosis, a neglected zoonotic reemerging disease. The immune response of hosts against these bacteria has not been thoroughly studied. Here, we studied over 6 months the antibody profiles in mice infected with *L. interrogans* and determined whether this humoral response confers long-term protection after homologous challenge six months after primary infection. Groups of mice were infected intraperitoneally with $2 \times 10^7$ bacteria of one of three different pathogenic serovars (Manilae, Copenhageni and Icterohaemorrhagiae) and some corresponding attenuated avirulent mutants. We measured by ELISA each type of *Leptospira*-specific immunoglobulin (Ig) (IgA, IgM, IgG and 4 subclasses) produced in the first weeks up to 6 months post-infection and studied their cross-reactivities among serovars. We showed different profiles of antibody response after *L. interrogans* challenge in mice, depending on the serovar and virulence of strains. However, all infected mice, including the ones harboring low antibody levels, like mice vaccinated with an inactivated, heat-killed strain, were protected against leptospirosis after challenge. Notably, we also showed an unusual sustained IgM response associated with chronic leptospiral colonization. Altogether, this long-term immune protection is different from what is known in humans and warrants further investigation.

## Introduction

Leptospirosis, a worldwide zoonosis, is a major public health problem with over one million cases and nearly 60,000 deaths annually [1,2]. It has a broad geographical distribution, but tropical/subtropical countries and resource-poor populations in developing countries carry the greatest burden [3–5]. Seasonal periods of heavy rainfall, flooding, and more globally, climate change also contribute to the risk of leptospirosis [6–10].

In developed countries disease outbreaks are usually associated with occupational exposure or leisure activities [11–14]. Infection occurs primarily after exposure to pathogenic leptospires excreted in the environment through the urine of asymptomatic animals, mostly rodents, considered as the reservoir [15]. Likewise, domestic animals and livestock can also become infected and develop leptospirosis.

In humans, clinical manifestations of leptospirosis are diverse, ranging from mild, self-limiting febrile illness to severe, life-threatening complications involving multi-organ dysfunctions such as Weil's syndrome [16]. The severity of the disease is related to the strain virulence and the inoculum dose, but also to the immune host response [17–19]. Antibiotic treatment is effective in most cases during the early stages of the infection, but untreated infections may lead to tissue damage, such as tubulo-interstitial nephritis [20].

The genus *Leptospira* is genetically highly heterogeneous, with 35 species currently recorded [21], including *L. interrogans*, the dominant pathogenic species worldwide. More than 350 serovars have been identified [22] based on the O-antigen portions of the lipopolysaccharide (LPS). This broad diversity has so far thwarted the development of a universal vaccine

against leptospirosis. Indeed, it has long been accepted that whole-leptospire-based vaccines confer only short-term immunity which is restricted to antigenically related serovars [23,24].

Experimental *Leptospira* infections of susceptible or resistant animal models have all reinforced the significant role of humoral immunity in protection against leptospirosis and pathogen clearance [25–28], with LPS as the main target to induce a protective antibody response [29]. Indeed, anti-leptospiral LPS-specific agglutinating antibodies are of prime importance in opsonizing leptospires, as demonstrated by *in vitro* experiments, or in conferring passive protection against experimental infection *in vivo* [26,28,30,31]. In humans, the presence of anti-*Leptospira* antibodies in patients who recovered from leptospirosis has been reported, as well as in subjects without known history of the disease, as a result of a natural asymptomatic infection [32]. However, very few studies have investigated whether this immunity persists over time and to what extent it is protective [33,34]. Partial information has arisen mostly from human [33] or hamster studies [34]. There is little information on the variability of humoral response according to the infecting *Leptospira* strain.

To overcome the serovar specificity barrier and, with the help of genomic, proteomic and metabolomic approaches, vaccine strategies based on recombinant or chimeric proteins defined as potential protective antigens have been developed and evaluated [24]. They demonstrate variable degrees of protection in animal models after *Leptospira* challenge, but fail to prevent renal colonization, which is a major issue in the control of leptospirosis [35–46]. Two attenuated mutants have recently been tested as vaccines. M895, a Manilae L495 mutant harbors a shorter, truncated LPS [47]. This mutant does not provoke any disease in the hamster model and does not colonize the mouse model. Another attenuated Copenhageni mutant, *fcpA*, is a motility-deficient mutant lacking the expression of a flagellar protein, FcpA [48]. Both have been used as cross-protective vaccines [44,49]. Single infection with either mutant protected mice and hamsters against homologous and heterologous challenges and reduced renal colonization [44,49]. Interestingly, it has been recently shown that the attenuated *fcp1/ fcpA* mutant conferred cross-protection through anti-*Leptospira* antibodies directed against proteins, but not through agglutinating antibodies. However, the isotypic nature of the immunoglobulins has not been investigated [49].

Whether in a natural or experimental leptospiral infection, analyses of antibody responses are crucial for diagnostics and determination of severity biomarkers [50], but also to provide insights into mechanisms of pathogenesis and to gain essential knowledge for the development of efficient vaccines. The lack of *in vitro* correlates of immunity against leptospirosis is a major hurdle. Indeed, determining new candidate antigens is useless if biological parameters to reliably assess the degree of protection against subsequent reinfection are not clearly established. Recent studies in mice with *L. interrrogans* Copenhageni Fiocruz L1-130 showed that experimental transdermal infection or through ocular and nasal mucosa, which are considered as natural routes of infection lead, like the classical intraperitoneal injection, to blood dissemination, and then renal colonization although with different kinetics [51,52]. In addition all routes of infection lead 15 days post-infection to comparable levels of total specific leptospires immunoglobulins [51,52].

Therefore, in this work, using our established mouse model of intraperitoneal infection, we established over 6 months the kinetics of antibody response and characterized the type and subclasses of anti-*Leptospira* immunoglobulins related to three pathogenic strains, representative of three distinct serovars and according to their viability, virulence and inoculum dose. Finally, we challenged immunized mice with a high dose of virulent leptospires and evaluated the level of protection.

## Materials and methods

### Ethics statement

The animal study was reviewed and approved under the protocol number dap190160 by the Institut Pasteur ethics committee (CETEA 89) (Paris, France), the competent authority, for compliance with the French and European (EU directive 2010/63) regulations on Animal Welfare and with Public Health Service recommendations. A novel scoring of clinical signs to establish critical points has been established for this study (S1C Fig).

### *Leptospira* strains and culture conditions

All strains of *Leptospira* used in this work were grown in Ellinghausen-McCullough-Johnson-Harris (EMJH) medium at 30˚C without agitation and subcultured weekly (twice a week for *L. biflexa*) to obtain late log phase cultures at the time of experiments. Three different serovars of *L. interrogans* (including wild-type strains, mutants, one derivative bioluminescent strain) and one serovar of saprophytic *L. biflexa* were used at different doses as listed in Table 1. *L. interrogans* Icterohaemorragiae strain Verdun, isolated in France from a soldier during World War I in 1917 in Verdun, was provided by the Centre National de Référence des leptospires (Institut Pasteur, France). It has been single-colony cloned (Cl3) after plating kidneys diluted in EMJH from a lethally infected young guinea pig, 5 days post-infection. This clone has been further weekly passaged in EMJH, at 30˚C, with agitation. Its virulence was assessed every 5 weeks in young guinea pigs by intraperitoneal infection with $2\times10^8$ bacteria in PBS. The virulence was assessed by death/critical end points reached within one week. Unexpectedly, this strain retained virulence for 2 years and then lost its ability to kill guinea pigs at the 104th weekly passage, which was confirmed by subsequent passages. This 2 year-passaged isolate (Cl3 p104) was thereafter designated Verdun AV. Whole genomic sequence data from the Craig Venter Institute are available for both strains (Verdun Cl3, low passage (LP) Genome ID: 1193019.3 | 31 Contigs; Verdun Cl3 p104 (AV), High Passage (HP) Genome ID: 1049910.3 | 1141 Contigs).

### Animals

Six-to eight-week old female or male C57BL/6J mice (Janvier Labs, France) were used in this study; they were randomly assigned to experimental groups (n = 5/group) and maintained in accordance with regulatory requirements throughout the experiments.

### Preparation of *Leptospira* and immunization

Leptospires were centrifuged, resuspended in endotoxin-free PBS (D-PBS Lonza), counted under dark field microscopy using a Petroff-Hauser chamber and diluted to the appropriate

**Table 1.** *Leptospira* strains used for immunization and/or challenge.

| *Leptospira* species Serovar | Strain | Immunization dose | Challenge dose | Characteristics | Reference |
|---|---|---|---|---|---|
| *L. interrogans* Manilae | L495 | $2\times10^7$ & $1\times10^5$ | | Wild-type (WT), Pathogenic | [47,53] |
| | M895 | $2\times10^7$ | | Truncated LPS | [47] |
| | HK L495 | $2\times10^7$ | | Heat-killed 100˚C 10 min | |
| | MFLum1 | NA | $5\times10^8$ | Bioluminescent WT | [54] |
| *L. interrogans* Copenhageni | Fiocruz LV2756 | $2\times10^7$ | $5\times10^8$ | WT, clinical isolate, Pathogenic | [48] |
| *L. interrogans* Icterohaemorrhagiae | Verdun Cl3 | $2\times10^7$ | $5\times10^8$ | Wild-type (WT), Pathogenic | [55] This study |
| | Verdun AV | $2\times10^7$ | | Highly passaged, Non pathogenic | This study |
| *L. biflexa* Patoc | Patoc 1 | $2\times10^7$ | | Saprophytic, Non pathogenic | [54] |

concentration. Bacterins were obtained by incubating the bacterial suspensions at 100˚C for 10 min. Bacterial inactivation was confirmed by culture for 2 weeks or absence of bioluminescence. For immunization, mice received a single dose of leptospires, in a volume of 200 µL, injected intraperitoneally (IP). Mice were immunized with either a sub-lethal dose ($2\times10^7$) of virulent strain or mutants or bacterins or a low dose ($1\times10^5$) of virulent strain. Mice immunized with *L. biflexa* strain Patoc 1 received a $2\times10^7$ dose.

## Blood sampling for serum collection

At designated time points (Days 3, 8, 15, 30, 60, 90, 120, 150, and 180 post-infection), blood samples (60 to maximum 80 µL) were collected by retromandibular puncture in serum-separating tubes (Microvette 200 serum-gel, Sarstedt TPP, Fisher Scientific). After centrifugation, serum was separated, aliquoted and stored at -20˚C until required.

## Challenge/protection assay

Six months after infection, mice were challenged with a high dose ($5\times10^8$ leptospires/mouse) of each virulent serovar, in a 200 µL final volume, administered IP. After infection, mice were monitored daily during the acute phase (up to 7/10 days), then weekly for weight loss and clinical signs of leptospirosis. Two months after challenge (unless indicated otherwise), mice were euthanized, kidneys were collected and stored at -80˚C for subsequent bacterial load quantification.

## Leptospiral load in kidneys

The presence of leptospires was determined in kidneys by quantitative real-time DNA PCR (qPCR). Total DNA was purified from approximately 25 mg of frozen tissue after mechanical disruption and using QiAamp DNA purification kit (Qiagen). The qPCR reaction was calibrated using a known number of heat-killed *L. interrogans*. DNA concentration was adjusted to around 200 ng in the qPCR reaction and normalized using the nidogen gene (encoding a protein of the glomerular basement membrane), as a calibrator of the cortex region of the kidney, with serial dilutions of known amounts of DNA. Using Primer Express 3 software, primers were designed for the unique *lpxA* gene of *L. interrogans* Fiocruz strain [56] to specifically detect pathogenic *Leptospira spp* (Forward (Fw): *5′-TTTTGCGTTTATTTCGGGACTT-3′*; Reverse primer (Rv): *5′-CAACCATTGAGTAATCTCCGACAA-3′*; Probe: *5′-TGCTGTACAT CAGTTTTG -3′*). Primers used to detect the nidogen gene used as a calibrator of cortex content were as described previously [57]. qPCR reactions were run on a Step one Plus real-time PCR apparatus using the absolute quantification program (Applied Biosystems), with the following conditions: 50˚C for 2 min, 95˚C for 10 min, followed by 40 cycles with denaturation at 95˚C for 15 s and annealing temperature 60˚C for 1 min, according to the manufacturer's instructions. Results of *lpxA* amplification were normalized with Nidogen housekeeping gene and expressed as the number of leptospires/200 ng of DNA. Detection limit of the method was set to 6 leptospires/200 ng of DNA.

## Determination and quantitation of anti-*Leptospira* immunoglobulins

Immunoglobulin types, subclasses and titers in immune and non-immune sera were measured by indirect homemade enzyme-linked immunosorbent assay (ELISA) with appropriate dilutions of serum collected at determined time-points after infection and tested in duplicate or triplicate, using *Leptospira* antigens (Ag) prepared as follows: leptospires from *L. interrogans* Manilae L495, Fiocruz LV2756, Verdun Cl3 or *L. biflexa* Patoc were grown to exponential

phase, washed in PBS, resuspended at $5 \times 10^8$ bacteria /mL and lysed using a FastPrep-24 Tissue and Cell Homogenizer in Lysing Matrix B tubes (MP Biomedicals) for 10 cycles with a setting of 4.0 m/s for 20 s. Unlike the heat inactivation-based protocol validated for the standard MAT agglutination test, this method preserves protein determinants and allows reactivity to all leptospiral antigens. The pH was adjusted to $\simeq 9.5$ with NaOH. Maxisorb 96-well plates (Nunc-immuno plates, Dutscher) were coated overnight at room temperature (RT) with 50 µL of *Leptospira* Ag (corresponding to 1300 to 1800 ng of protein per well), blocked with 100 µL of PBS-10% SVF for 1 h at RT. Serum samples were diluted in blocking buffer, added into corresponding wells (50 µL/well) and incubated for 2 h at RT. Immunoglobulins were detected with horseradish peroxidase (HRP) goat anti-mouse Ig (BD Pharmingen, 1/2000 diluted), IgM, IgG, IgA, IgG1, IgG3, IgG2b, or IgG2c antibodies (all from Abcam, 1/5000 or 1/10000 diluted), added to the plates and incubated at RT for 1 h. Washing with PBST (PBS with 0.05% [v/v] Tween 20 %) was performed three times between all steps. The peroxidase activity was measured with TMB substrate (Sigma-Aldrich), and stopped after 20 min incubation with 1N HCL. Optical density (OD) was read at 450 nm on an Elx808 Microplate reader (BioTek) and mean values were obtained from serum samples assayed in duplicate or triplicate. In each experiment, non-specific reactivity of HRP-labelled specific anti-mouse Ig was checked on leptospiral antigens. OD values were all in the range of 0,05–0,07 considered as the background of the assay. Data were expressed as mean values of duplicates or triplicates, subtracted from the background value.

## Results

### Differential production of specific immunoglobulins according to *Leptospira* strains

As B cell-mediated humoral immune responses are critical for host defense against leptospirosis [28], we thought to determine the anti-*Leptospira* immunoglobulin (Ig) profiles elicited over time after experimental infection in our model of C57BL6/J mice. We used three virulent strains of pathogenic *L. interrogans*: Manilae L495, Fiocruz LV2756 and Verdun Cl3, respectively representative of serovar Manilae, Copenhageni and Icterohaemorrhagiæ, and one saprophytic strain of *L. biflexa*, serovar Patoc strain Patoc 1 (Table 1).

Mice were injected intraperitoneally with a dose of $2 \times 10^7$ live leptospires, weighed individually and blood sampled at defined time-points post-infection (p.i.) to observe and measure the early production (day (D)3, D8 and D15) of anti-*Leptospira* immunoglobulins (Igs) in serum and their evolution (D30, D60, D90 and D120) and long-term persistence up to 6 months p.i. (D180). Compared to their initial weight, mice injected with the virulent strains lost a maximum of 3.5%, 9.1% and 9.3%, respectively for Icterohaemorrhagiae Verdun, Manilae L495 and Fiocruz LV2756 groups during the acute phase (D1 to D3 p.i.) (S1A Fig). This weight loss was quickly compensated in 3 to 5 days. As expected, no severe clinical signs were recorded after infection at this dose of $2 \times 10^7$ leptospires (S1B Fig).

We first measured the kinetics of total anti-*Leptospira* Igs in individual or pooled sera collected per time-point (Fig 1A). As expected, no anti-*Leptospira* response was detected in control groups injected with PBS. In infected mice, we found specific *Leptospira* Igs, as early as 3 days p.i., with levels increasing to a maximum around D30 p.i. for Verdun Cl3, D60 p.i. for Manilae L495 and D90 p.i. for Fiocruz LV2756 (Fig 1A). Then, the antibody levels slowly decreased over time to reach at D180 about half the levels of their maximum titers. Notably, Ig titers were 2- to 4 -fold higher for the Manilae L495 or Fiocruz LV2756 strains than in mice injected with the Verdun Cl3 strain at D180, although until D30 the kinetics and profiles of specific Igs were rather similar between the 3 virulent strains (Fig 1A). In contrast, levels of

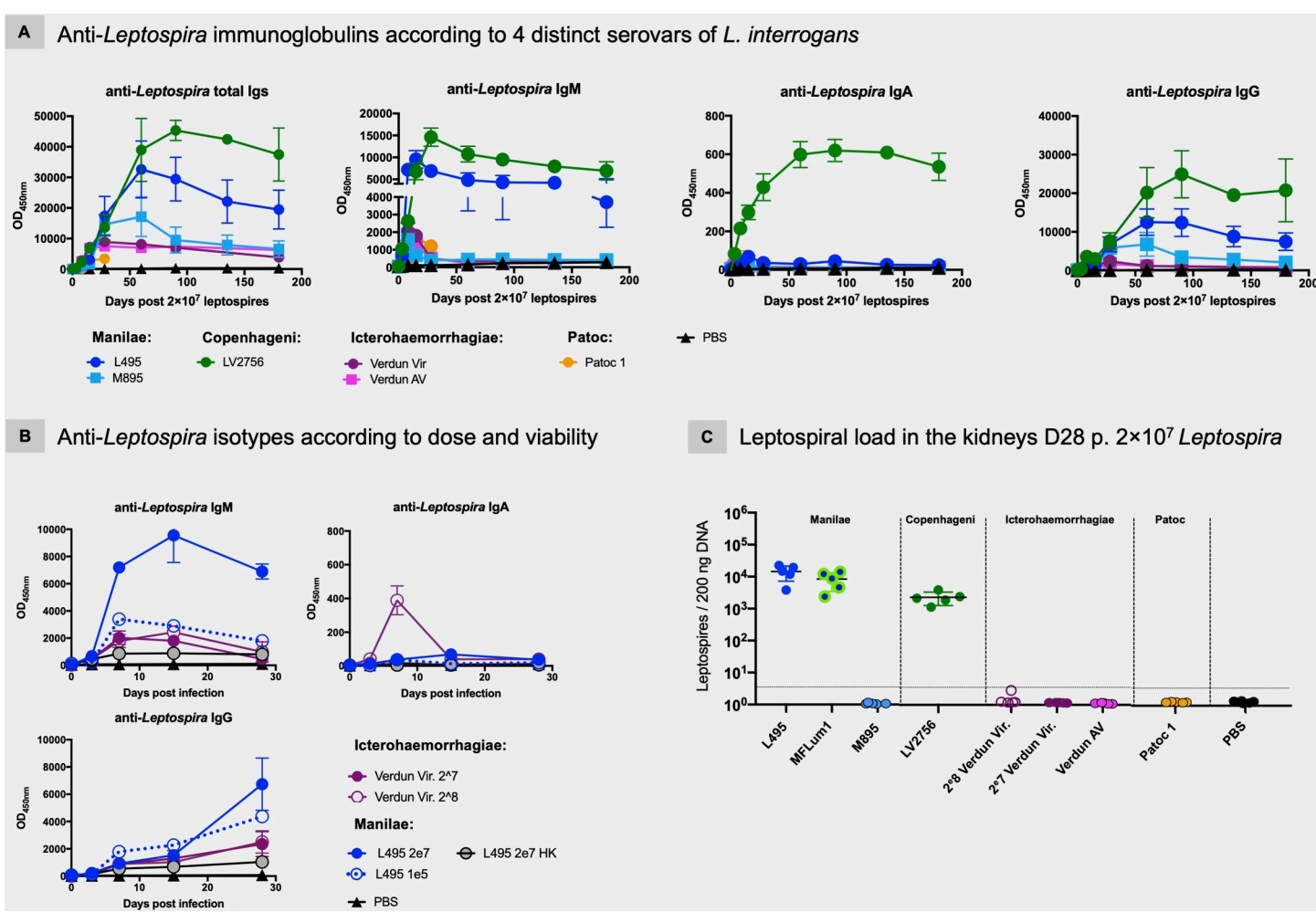

**Fig 1. Sustained and persistent specific antibodies, including IgM that correlate with presence of leptospires in kidneys.** Anti-*Leptospira* immunoglobulins generated after experimental infection of C57BL/6J mice with representative strains of four distinct serovars (Manilae, Copenhageni, Icterohaemorrhagiae and Patoc) of leptospires. A) Kinetics and isotyping determination over a six-month period (28 days for Patoc) after infection. Mice were intraperitoneally inoculated with $2 \times 10^7$ of virulent Manilae L495 strain (dark blue circle) or avirulent Manilae M895 mutant (light blue square), or virulent Copenhageni Fiocruz LV2756 strain (dark green circle) or virulent Icterohaemorrhagiae Verdun strain (dark pink circle) or avirulent (AV) strain (light pink square) or saprophytic Patoc Patoc 1 strain (orange circle), or PBS as negative control. B) Profiles of 3 specific different isotypes (IgM, IgA, IgG) produced over the first month after experimental infection with different doses of virulent strains (Icterohaemorrhagiae Verdun or Manilae L495 strains) or inactivated heat killed (HK) Manilae L495 strain. Mice were intraperitoneally inoculated with $2 \times 10^8$ (empty pink circle) or with $2 \times 10^7$ (dark pink circle) of virulent Icterohaemorrhagiae Verdun, or with $1 \times 10^5$ of virulent Manilae L495 strain (dashed blue line) or $2 \times 10^7$ of heat-inactivated L495 (grey circle) or $2 \times 10^7$ (dark blue circle) of virulent Manilae L495, or with PBS as negative control. Each figure represents the profiles for total Igs or each specific isotypes obtained from serum tested in pool, for a same experimental group, at D0, D3 post-infection (p.i.) or individually (when error bars), each dot being a determined time-point p.i. Absence of visible error bars post D8 p.i., notably for Icterohaemorrhagiae and Patoc patterns performed with individual samples, is due to the scale. Specific Ig responses were assessed up to day 180 p.i. with female mice (n = 5/group) or day 28 p.i. with male mice (n = 5/group). C) Leptospiral loads in kidneys of mice 28 days after intraperitoneal infection with representative strains of three distinct pathogenic (Manilae, Copenhageni, Icterohaemorrhagiae) and one saprophytic (Patoc) serovars. Mice were inoculated with $2 \times 10^7$ of virulent Manilae L495 or bioluminescent Manilae L495 MFLum1 derivative strains or avirulent Manilae M895 mutant, or virulent Copenhageni Fiocruz LV2756 strain or virulent Icterohaemorrhagiae Verdun strain (2 doses) or avirulent (AV) strain, or saprophytic Patoc 1 strain (serovar Patoc), or PBS as negative control. After euthanasia at D28 p.i., kidneys were collected and DNA purification performed before qPCR amplification. Each dot corresponds to one sample, n = 5/group.

antibodies elicited up to D28 after injection of the saprophytic Patoc 1 strain were very low (Fig 1A). In general, over the first month following infection, we did not observe any major differences related to the sex of mice with either the kinetics or the level of Igs produced (S1D Fig). In conclusion, the production of antibodies was concomitant with clinical manifestations of the disease, when they occurred.

## Virulence effect in anti-*Leptospira* immunoglobulin production

Because of the striking difference in Ig profiles between the *L. interrogans* and *L. biflexa* strains, we further aimed to determine if the virulence could affect the development and the evolution of the humoral response. Therefore, we also measured the total anti-*Leptospira* Igs in serum of mice inoculated with avirulent mutants of the Manilae and Icterohaemorragiae strains (Fig 1A). The Manilae M895 mutant has a truncated LPS [47]. Like the highly passaged Verdun AV strain that lost its ability to kill young guinea pigs, the attenuated M895 has lost its ability to cause disease in the standard hamster model of leptospirosis [58]. Infections with these two strains resulted in lower clinical manifestations (S1A and S1B Fig), showing that they are also attenuated in the mouse model. The M895 mutant elicited similar kinetics and levels of specific anti-*Leptospira* Igs until D60 p.i., but thereafter showed lower levels than those obtained after inoculation with the virulent strain (Fig 1A). In contrast, the Ig response obtained in mice inoculated with the Icterohaemorrhagiae avirulent Verdun strain (Fig 1A) appeared identical to the Ig profile obtained with the virulent strain, indicating that the 2 years of *in vitro* passages did not affect the immunogenicity of the Verdun strain. Notably, the total Ig profiles elicited by both Icterohaemorrhagiae Verdun strains resemble more closely the response to avirulent Manilae M895, than to its virulent L495 counterpart. These results suggest that the amplitude and persistence of the anti-*Leptospira* humoral response in the C57BL6/J mouse model depend on the virulence but also on serovars, with a major difference with the Verdun compared to the Manilae L495 or Fiocruz LV2756 strains.

## Isotyping and persistence of the different anti-*Leptospira* immunoglobulins

Next, to gain a better insight in the humoral response, we measured the kinetics of each anti-*Leptospira* isotype up to 6 months post-infection (Fig 1A). The IgM antibodies were the first to appear, at D3 p.i., regardless of strain. However, the magnitude and profiles of IgM production were distinct and appeared to be related to the serovar and virulence of the strains (Fig 1A). The IgM responses elicited by the avirulent mutant strains showed, as expected, a maximum titer measured at D8 p.i. that decreased thereafter. However, the IgM production profiles for both Manilae L495 and Fiocruz LV2756 strains were strikingly different. Indeed, for both, IgM titers progressed to reach a peak at D15 or D30 p.i., and decreased until D60 p.i. but then surprisingly persisted at sustained high levels (Fig 1A). On the other hand, the low IgM response elicited after infection with the virulent Icterohaemorrhagiae Verdun strain showed a typical IgM profile, although with a peak of production at D15 p.i. (Fig 1A).

Anti-*Leptospira* IgA antibodies were detected barely or not at all after infection with the Icterohaemorrhagiae Verdun strains and only a weak and transient IgA response was detected after inoculation with Manilae L495 or M895. In striking contrast, the virulent Copenhageni Fiocruz LV2756 strain elicited a sustained IgA response (Fig 1A).

Secondarily to IgM antibodies, the different *Leptospira* inoculations elicited specific IgG antibodies, with increasing titers detected up to D60-D90 p.i., depending on the serovar tested (Fig 1A, right panel). Similarly to the results for total Ig, while both Manilae strains and the Fiocruz LV2756 induced a high and sustained IgG response, we observed that both Icterohaemorrhagiae strains generated equivalent IgG responses of lower magnitude. Mice inoculated with the saprophytic Patoc 1 strain also elicited very low levels of specific IgG.

In conclusion, these results showed that the virulent Manilae L495 and Copenhageni LV2756 strains primarily generated a robust and sustained specific IgM response, supported by a long-term IgG response. Conversely, with an equivalent infecting dose, the virulent Icterohaemorrhagiae Verdun strain and saprophytic Patoc 1 strain elicited rather low-level humoral responses. Overall, these data confirmed antigenic diversity between *Leptospira*

strains and differences in host recognition impacting the development of humoral immune responses, including the IgA response.

## Inoculum effects on the different anti-*Leptospira* isotypes

Because the virulence of the three different pathogenic strains was not the same, we sought to determine whether a higher dose of the Icterohaemorrhagiae Verdun strain could lead to Ig profiles similar to those obtained with the Manilae L495 and Copenhageni Fiocruz LV2756 strains. Therefore, we examined the kinetics of production of anti-*Leptospira* Igs with a 10-fold higher ($2\times10^8$) dose of Icterohaemorrhagiae strain. Under these conditions, mice showed severe clinical signs with weakness, jaundice and an average loss of 16.4% of their initial weight at D5 p.i., before they recovered, demonstrating the virulent nature of this strain in our mouse model (S1A Fig). Interestingly, the specific anti-*Leptospira* (Icterohaemorrhagiae) IgM and IgG profiles appeared quite similar in terms of kinetics, but more surprisingly, also in terms of magnitude compared to the titers obtained with the $2\times10^7$ leptospires/mouse dose (Fig 1B). However, with the higher dose of $2\times10^8$, a transient IgA response was observed, peaking at D8 p.i.

To assess whether the inoculum dose could also impact the production of the anti-Manilae Igs, we also quantified the specific isotypes generated after inoculation of a low dose ($1\times10^5$ leptospires) of Manilae L495. In contrast to the persistent response observed in mice injected with $2\times10^7$ leptospires, the IgM response generated with the low dose peaked at D8 p.i., and dropped continuously, although the IgG responses appeared roughly similar in terms of kinetics, magnitude and persistence (Fig 1B). The IgA responses elicited with the low dose of Manilae were close to background levels (Fig 1B).

Altogether, these results suggest isotype differences in the anti-*Leptospira* humoral response according to the infectious dose, indicating a differential host response or threshold level to trigger the host immune system.

## Minimal specific antibody response after inoculation with inactivated leptospires

Next, to determine whether the humoral responses observed required live bacteria, we determined the Ig responses in the sera of mice injected with $2\times10^7$ heat-killed Manilae L495 (L495 HK), inactivated by a 10 min heating at 100°C (Fig 1B). The levels of IgM, IgA and IgG detected were much lower than in any other condition of inoculation with L495 strain, and close to the background level for IgA. These results suggest that infection with live bacteria elicits a completely different antibody response compared to injection of the heat-killed bacterin.

## *Leptospira* loads in kidney after a sub-lethal dose infection

In order to determine any relationship between renal colonization and host humoral response in infected mice, we quantified the leptospiral load in kidneys of mice inoculated with the same dose of $2\times10^7$ leptospires. Mice were euthanized 28 days post-infection, when renal colonization was reliably established [54]. We quantified high and equivalent levels of leptospiral DNA in both groups infected with Manilae L495 and MFLum1, the bioluminescent derivative of L495 [54]. We also found high levels of bacteria in kidneys of mice infected with Copenhageni Fiocruz LV2756 strain (Fig 1C), although about one log less in magnitude compared with the Manilae strains. Interestingly, after infection with a similar dose of virulent Icterohaemorrhagiae Verdun Cl3, we did not find leptospiral DNA in the kidneys. In addition, detection of the Icterohaemorrhagiae Verdun Cl3 inoculated in a 10 fold-higher dose was also negative in 4

out of 5 mice, and barely detected in the last one, suggesting that this strain does not colonize the mouse kidney. As expected, we did not detect leptospires in the kidneys of any of the mice inoculated with the Manilae M895 avirulent mutant, nor in mice inoculated with the Patoc 1 strain, or PBS (Fig 1C). These results demonstrate major pathophysiological differences between the three pathogenic *L. interrogans* serovars.

## Determination of the different subclasses of anti-*Leptospira* IgG antibodies

Since we observed variations in kinetics and magnitude of specific IgG responses, we further characterized the different specific IgG subclasses (IgG1, IgG2b, IgG2c and IgG3) to examine whether an antibody response profile could be associated with a specific serovar, notably in terms of long-term immunity (Fig 2).

Both Manilae L495 and Copenhageni Fiocruz LV2756 elicited similar profiles of IgG1 and IgG3, peaking at D60 p.i. and decreasing to reach between a third and a half of their maximum at D180 p.i. For the determination of IgG2 subclasses, we assessed the production of specific anti-*Leptospira* IgG2b and IgG2c. The inbred mouse strain C57BL/6 with the Igh1-b allele does not have the gene for IgG2a and instead expresses the IgG2c isotype [59]. Both IgG2b

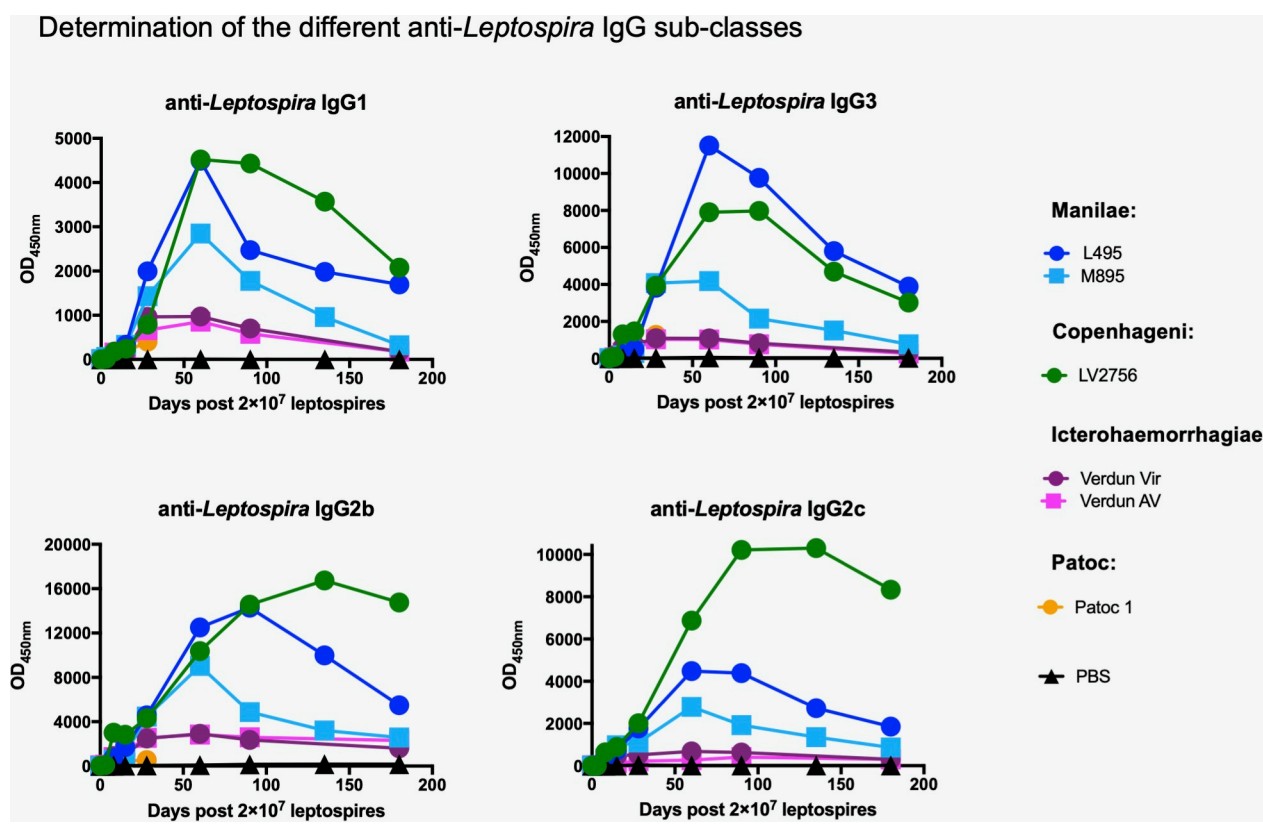

**Fig 2. Determination of IgG subclasses reveals antigenic diversity between *L. interrogans* strains.** Profiles of four anti-*Leptospira* IgG subclasses generated over time after experimental infection with representative strains of three distinct pathogenic (Manilae, Copenhageni and Icterohaemorrhagiae) and one saprophytic (Patoc) serovars. Specific IgG1, IgG3 (upper panels) and IgG2b and IgG2c (lower panels) antibodies produced over 6 months in mice inoculated with $2\times10^7$ of virulent Manilae L495 strain (dark blue circle) or avirulent Manilae M895 mutant (light blue square), or virulent Copenhageni Fiocruz LV2756 strain (dark green circle), or virulent Icterohaemorrhagiae Verdun strain (dark pink circle) or avirulent (AV) strain (light pink square), or Patoc 1 strain (orange circle), or PBS as negative control. Specific IgG subclasses were determined by ELISA using specific *Leptospira* antigen preparation and appropriate dilutions of serum collected at determined time-point after infection. Each figure represents the profiles for each specific IgG subclass obtained from serum tested in pool (for a same experimental group). Antibody responses were assessed up to day 180 p.i. with female mice (n = 5).

and IgG2c isotypes were elicited by infection with Manilae L495 and Copenhageni Fiocruz LV2756. IgG2b and IgG2c profiles elicited by L495 were similar to the other IgG subclasses, with responses decreasing to a third of their maximum at D180 p.i. Fiocruz LV2756 induced higher responses, with peaks between D90 p.i. and D120 p.i., and a high sustained response at D180 p.i. Infections with the Manilae M895 mutant elicited all four IgG subclasses, but in lower magnitude compared to their virulent counterparts. Consistent with the anti-*Leptospira* IgG responses, the two Icterohaemorrhagiae Verdun strains elicited identical profiles of the four different IgG subclasses, although with a lower magnitude compared to the Manilae L495 and Copenhageni Fiocruz LV2756 strains. The Patoc 1 strain elicited only a very low IgG response, and thus, weak IgG subclass responses (Fig 2).

## Cross-reactive anti-*Leptospira* immunoglobulins in sera of pre-infected mice

Humoral immunity against *Leptospira* following infection is mainly described as LPS-dependent and consequently as serovar-specific. However, cross-reactions against related serovars have been reported [49,60]. We assessed the cross-reactivity of anti-*Leptospira* immunoglobulins obtained after pre-infection with the virulent strain of three distinct serovars against the two other serovars'antigens.

While specific immunoglobulins obtained after Manilae L495 or Verdun infection did not show strong evidence of reactivity with the other serovars, we unexpectedly observed that IgM, IgA and IgG antibodies resulting from the Copenhageni Fiocruz LV2756 infection cross-reacted strongly with Icterohaemorrhagiae Verdun antigen and IgG, with Manilae L495 antigen (Fig 3). Accordingly, all IgG four subclasses elicited after Fiocruz infection strongly cross-reacted and recognized Verdun antigens, and to a lesser extent Manilae antigens. Interestingly, while IgG1 and IgG3 elicited by Manilae infection barely cross-reacted, IgG2 (b and c) against Manilae cross-reacted well with both Verdun and Fiocruz antigens, although the Ig subclasses elicited after Verdun infection did not cross-react with Fiocruz, nor with Manilae (S2 Fig). These results show that the low homologous humoral responses observed after the Verdun infection are not due to a technical issue in antigen preparation, and rather suggest a specific ability of the Icterohaemorrhagiae Verdun strain to avoid the development of a strong immunoglobulin response. In addition, the results also suggest the existence of a complex immune response after *Leptospira* infection with some serovars able to confer cross-reactivity. Interestingly, we evidenced that the cross-reactivity conferred by one serovar against another is not necessarily a reciprocal relationship.

## Long-term protective effect of pre-infection

To evaluate the protective capacity of the long-term humoral immunity, we challenged the mice at D180 p.i. with a high dose ($5 \times 10^8$ leptospires) of the corresponding virulent parent strain. Controls included mice that were initially injected with PBS at D0, and then challenged at D180. After the challenge, individual weight variation (Fig 4A) and clinical signs of the disease were registered daily during the acute phase (S1B Fig), then weekly, according to general and specific criteria. As expected, the challenged control mice showed clinical signs of the disease, of high severity after Manilae MFLum1 and Icterohaemorrhagiae Verdun challenges, less intense after Copenhageni Fiocruz LV2756 (Figs 4A and S1B). Conversely, all pre-infected mice, regardless of the strain, live or heat killed, virulent or attenuated, showed few clinical signs. Hence, in mice pre-infected with either Manilae L495 or M895 strains, the challenge caused only a weak disease as evidenced by the severity score that never exceeded the value of 3 during the acute phase (S1B Fig). Mice pre-infected with Manilae L495 lost a maximum of

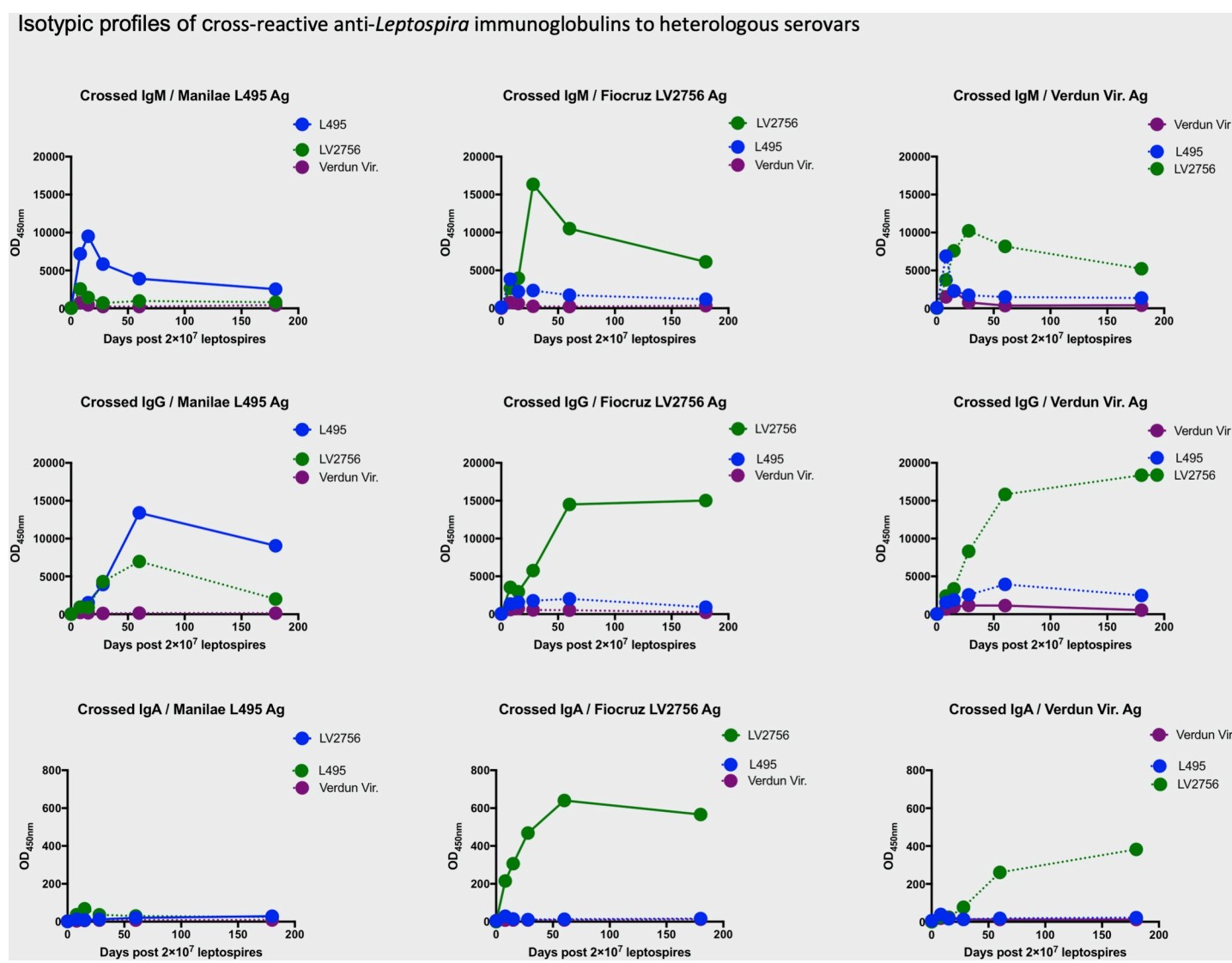

**Fig 3. Infection with Fiocruz LV2756 virulent strain leads to cross-reactive antibodies.** Profiles of different isotypes of anti-*Leptospira* immunoglobulins tested on heterologous serovars. Specific IgM (top row), IgG (central row) and IgA (bottom row) immunoglobulins obtained after experimental infection with $2\times10^{7}$ virulent leptospires representative of 3 distinct serovars were checked against heterologous serovar antigen preparation (dashed line) and compared to the profile obtained with homologous serovar (full line). Anti-*Leptospira* Ig isotypes were determined by ELISA assay using specific *Leptospira* antigen preparations Manilae L495 (left column), or Copenhageni Fiocruz LV2756 (central column) or Icterohaemorrhagiae Verdun (right column) and appropriate dilutions of serum collected at determined time-point after infection. Each figure represents the profiles for specific IgM, IgG and IgA isotypes obtained from serum tested in pool (for a same experimental group). Antibody responses were assessed up to day 180 p.i. with female mice (n = 5).

6% of their initial weight, stably and continuously for 14 days before recovering slightly (Fig 4A, left panel), except one mouse that required euthanasia 16 days post-challenge (p.C.). Mice immunized with Manilae M895 or a low dose of Manilae L495 lost around 5.5% only at D1 p. C., and then subsequently compensated, whereas mice initially injected with heat-killed L495 presented more pronounced signs before later recovering. For mice infected with Manilae strains, we used the bioluminescent MFLum1 strain which allowed us to monitor the infection course in real time by *in vivo* imaging (IVIS) [54] (S3A Fig). IVIS performed at different time-points on all mice included in the Manilae groups, showed at least a 100-fold difference between the light emitted, reflecting the levels of live leptospires, that were as soon as D1 p.i. high in controls, compared to background levels in pre-infected mice, regardless of the

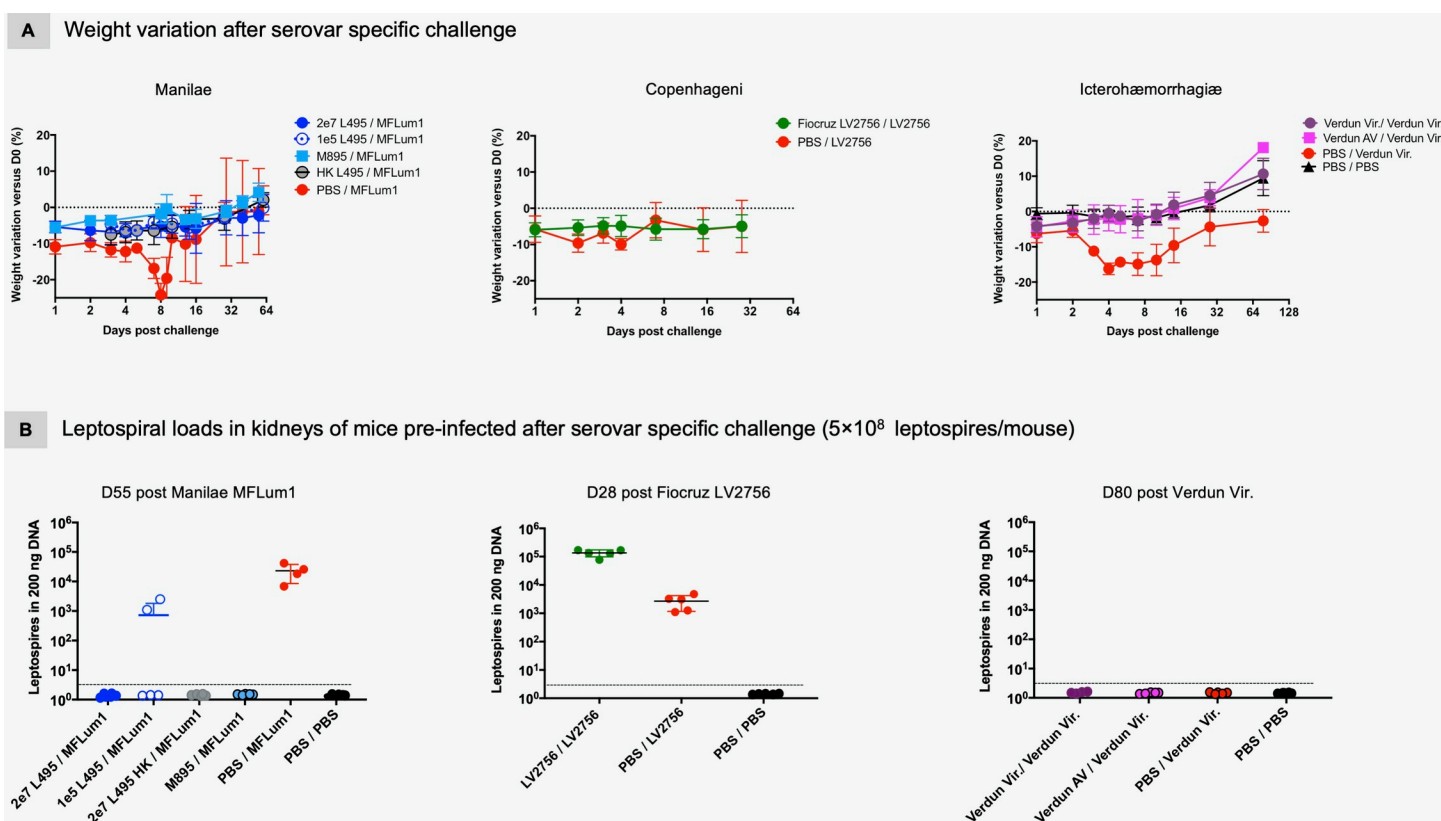

**Fig 4. Pre-infection elicits a serovar long-term protective immunity against subsequent homologous *Leptospira* challenge.** Outcome evaluation after serovar specific challenge in mice 6 months after *L. interrogans* pre-infection. A) Weight evolution individually recorded in mice pre-infected with different doses of Manilae L495, or M895 mutant or inactivated L495, and challenged with 5×10⁸ of Manilae bioluminescent MFLum1 strain or pre-infected with Copenhageni Fiocruz LV2756 strain and challenged with 5×10⁸ of Fiocruz LV2756 strain, or pre-infected with virulent Icterohaemorrhagiae Verdun strain or avirulent (AV) strain and challenged with 5×10⁸ of virulent Verdun strain. Control corresponds to mice initially injected with PBS and challenged with the respective virulent strain specific to each experimental condition. PBS/PBS (in Verdun groups) corresponds to mice injected with PBS instead of leptospires challenge as negative control. Weight change from the initial weight at the day of infection, expressed as a percentage, was daily recorded during the acute phase of the disease (D0 to D7 p.i.) then weekly (D8 to the end of the experiment). Graphs represent the mean ± SD of the weight change recorded overtime after challenge for each experimental group (n = 5/group). B) Leptospiral loads in kidneys of mice pre-infected for 6 months and then serovar specifically challenged with 5×10⁸ leptospires/mouse. Mice initially pre-infected with 2 different doses of virulent Manilae L495, or heat-killed L495 or avirulent M895 mutant and challenged with Manilae derivative MFLum1, or pre-infected and challenged with virulent Fiocruz LV2756 strain, or pre-infected with virulent Icterohaemorrhagiae Verdun strain or avirulent (AV) strain and challenged with virulent Verdun were euthanized when indicated. Kidneys were collected and the bacterial loads were measured by qPCR. Control corresponds to mice initially injected with PBS and challenged with the respective virulent strain specific to each experimental condition. PBS corresponds to mice injected with PBS as negative control. Results were normalized with Nidogen housekeeping gene and expressed as the number of leptospires/200 ng of DNA. Each dot corresponds to one sample, n = 5/group with exception of control Manilae group (n = 4; one mouse was euthanased after challenge).

immunization condition (S3A Fig). For mice pre-infected with Copenhageni Fiocruz LV2756, clinical signs of the disease after the high dose homologous challenge were significantly different from those observed in controls, although for only 2 days, with a 2-fold less weight loss than mice in the control group and a clinical severity score that never exceeded the 2 value (Fig 4A, central panel). With Icterohaemorrhagiae, differences in terms of clinical outcome between pre-infected mice and controls after the challenge were more pronounced. Indeed, while controls showed severe signs of the disease with high weight loss and high individual severity score (S1A and S1B Fig), both groups of pre-infected mice lost 4-fold less weight during the acute phase before quickly recovering (Fig 4A, right panel). Altogether, these results showed that a pre-infection 6 months before challenge, regardless of the serovar/strain of leptospires and whether the strain was virulent or not, stimulated a protective immune response, at least when challenged with a high dose of the same virulent strain.

## Impact of pre-infection on the leptospiral burden in kidneys after challenge

Since we demonstrated that pre-infected mice were immune and so able to withstand further homologous infection, we evaluated the extent to which this protection allowed the host to limit the infection, notably on leptospiral burden in kidneys. Pre-infected mice and control mice were euthanized after challenge and kidneys collected to perform qPCR quantification of leptospiral DNA. As expected, control mice challenged with a high dose of Manilae MFLum1 showed renal colonization, whereas no leptospires were detected in the kidneys of mice pre-infected with attenuated or heat-killed Manilae L495, showing that the corresponding immune responses were totally protective and blocked the renal colonization. However, surprisingly, no leptospires were detected in the kidneys of mice pre-infected with $2 \times 10^7$ live L495, although two mice initially inoculated with the low dose ($1 \times 10^5$ leptospires) of Manilae L495 presented a residual leptospiral load (Fig 4B, left panel). The results appeared strikingly different in mice pre-infected with the Copenhageni Fiocruz LV2756 strain. Compared to the loads measured in controls, we observed a higher quantity of leptospires in Copenhageni pre-infected mice, suggesting that leptospires present in the kidneys following the primary infection could accumulate with those of the subsequent infection (Fig 4B, central panel). These results suggest that pre-infection with this Copenhageni strain conferred to the host immune system a partial protective capacity against further acute infection, but it was not able to prevent renal colonization. Concerning groups of mice pre-infected with the virulent or avirulent Icterohaemorrhagiae Verdun strains, two months after challenge, there was no leptospiral DNA in the kidneys of mice (Fig 4B, right panel). Moreover, we did not find leptospires in the kidneys of control mice, supporting the previous results obtained at D28 p.i. and confirming that the virulent Icterohaemorrhagiae Verdun strain does not induce persistent renal colonization in C57BL/6J mice. Altogether, these data demonstrated that, although pre-infection by leptospires confers evident benefit in terms of immunity against acute leptospirosis in the mouse model, the elicited protection against renal colonization is differential and the protective efficiency depends on the infecting strain. In addition, our results suggest that the virulent nature of the strain alone would not explain these differences in terms of protection, suggesting that the recognition/signaling pathways to induce humoral immunity in the host may vary according to the *Leptospira* serovar and/or strain.

## Magnitude of the specific anti-*Leptospira* response after challenge

To determine the antibody responses following challenge, we collected serum at different time-points post-challenge (p.C.) and performed ELISA to assess specific anti-*Leptospira* Igs (Fig 5). Eight days p.C., we observed an increase in IgM responses in all groups of mice but to a higher extent in immunized (pre-infected) mice compared to controls, regardless of the challenge strain or the dose (Fig 5A, upper panels and S3B Fig). The IgM boost observed after the challenge with Icterohaemorrhagiae Verdun was very low compared to Manilae MFLum1 and Copenhageni Fiocruz LV2756 challenges. IgM titers in control mice (PBS/challenged) were essentially similar to those measured after the first infection, respectively to each virulent strain tested, and with similar production kinetics.

In contrast to the primary infection at the dose of $2 \times 10^7$ with the Copenhageni or Manilae strains that elicited either sustained IgA or did not elicit IgA, the infection with a high dose of bacteria used as challenge triggered in control mice a specific transient IgA response, peaking at D8 p.i. in all groups (Fig 5A, lower panels). This suggests a probable dose effect, as already observed for the virulent Icterohaemorrhagiae Verdun strain (Fig 1B). Notably, the IgA response elicited in the groups immunized with the Manilae L495 and Copenhageni Fiocruz LV2756 strains was very high. An IgA boost of lower intensity was also observed with the low

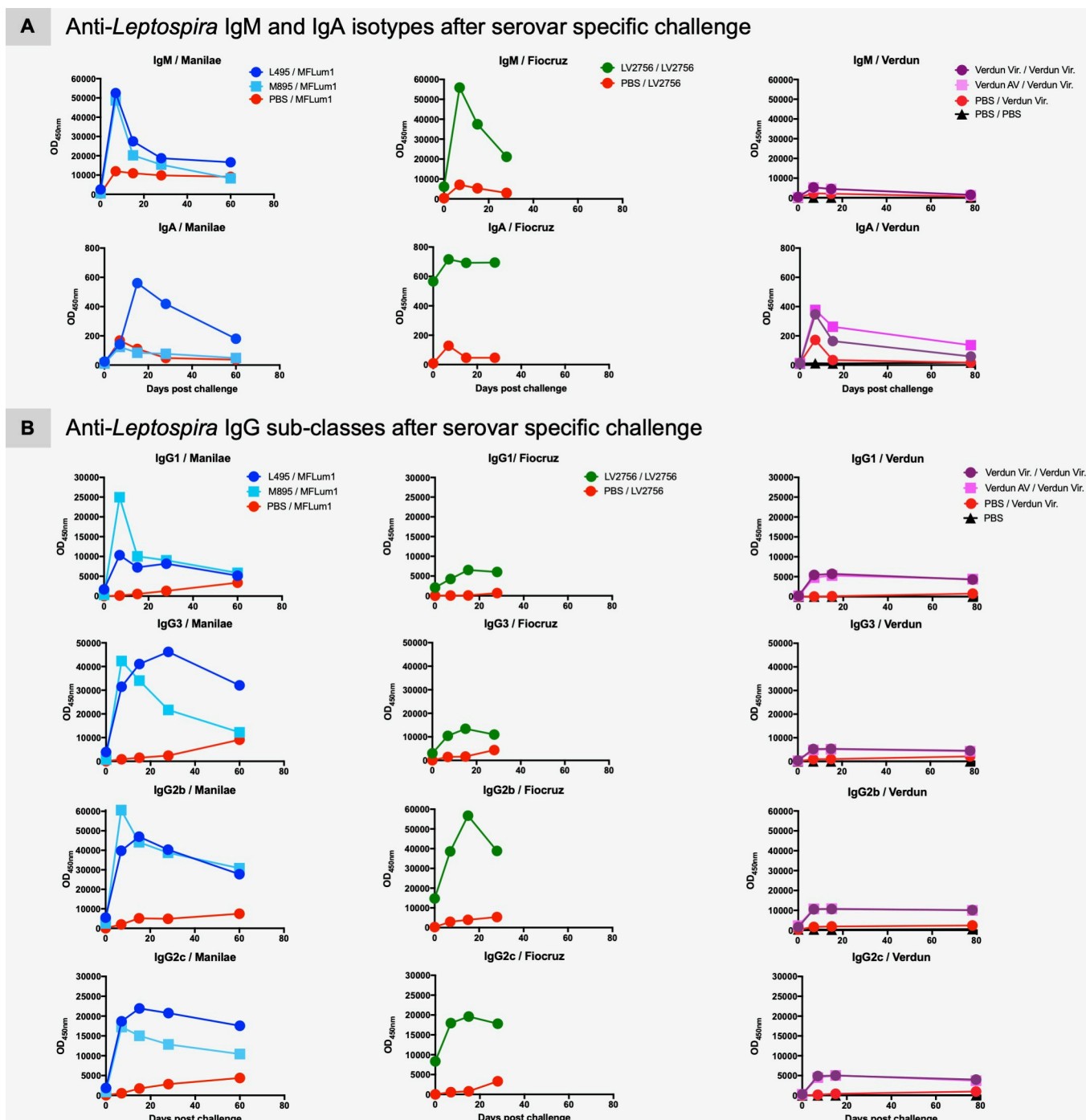

**Fig 5. Homologous challenge boosts the specific anti-*Leptospira* humoral immunity including IgG1 and IgG3 responses.** A) Profiles of anti-*Leptospira* IgM and IgG antibodies after serovar specific challenge in mice immunized with *L. interrogans* Manilae or Copenhageni or Icterohaemorrhagiae strains. Specific IgM (upper panels) and specific IgA (lower panels) produced after Manilae bioluminescent derivative MFLum1 challenge in mice initially immunized with Manilae L495 or M895 (left panels), or after homologous challenge in mice initially immunized with Copenhageni Fiocruz LV2756 (central panels), or after Icterohaemorrhagiae Verdun challenge in mice immunized with virulent Verdun or avirulent (AV) (right panels). B) Profiles of four specific anti-*Leptospira* IgG subclasses elicited after serovar specific challenge in mice immunized with *L. interrogans* Manilae or Copenhageni or Icterohaemorrhagiae strains. IgG1, IgG3, IgG2b, IgG2c antibodies produced after Manilae bioluminescent derivative MFLum1 challenge in mice initially immunized with Manilae L495 or M895 strains (left panels), or after homologous challenge in mice initially immunized with Fiocruz LV2756 (central panels), or after Icterohaemorrhagiae Verdun challenge in mice immunized with virulent Verdun or avirulent (right panels). Control corresponds to mice initially injected with PBS and challenged with the respective virulent strain specific to each experimental condition and PBS/PBS (in Verdun groups) corresponds to mice injected with PBS instead pre-infection and challenge as negative control. Each figure represents the profiles for IgM and IgA isotypes and IgG subclasses obtained from serum tested in pool (for a same

experimental group). Antibody responses were assessed up to D55 p.C. (Manilae groups), D28 p.C. (Copenhageni groups) and D78 p.C. (Icterohaemorrhagiae groups) with n = 5 female mice/group with exception of control Manilae group (n = 4; one mouse died after challenge).

dose of Manilae L495 and heat-killed L495 (S3B Fig), but in a lower extent with the attenuated Manilae M895 mutant (Fig 5A, lower panels).

Compared to control mice, the IgG responses of all groups of pre-infected mice were strikingly boosted after specific challenge (Fig 5B). Indeed, a boost in levels of all four IgG subclasses was observed in all mice immunized and challenged with all serovars, with sustained high levels one or two months p.C. A higher boost of IgG1, IgG2b and IgG3 antibodies was observed in M895 pre-infected in mice compared to the Manilae L495 pre-infected ones (Fig 5B, left panels). Higher levels of IgG subclasses were also observed in mice pre-infected with the Copenhageni strain (Fig 5B, central panels). We also interestingly observed that all IgG subclasses in Icterohaemorrhagiae immunized mice appeared much higher than expected in comparison with the rather low response generated after pre-infection, or in control mice (Fig 5B, right panels). Interestingly, immunization with heat-killed Manilae L495 induced a boost in IgG1 and IgG3, with a sustained response one month p.C. IgG2c, not IgG2b, were also boosted but only in the first days p.C. (S3C Fig).

## Discussion

It has long been accepted that immunity against leptospirosis is based on the humoral response, considered as short term in humans and animals, but it is not clear whether this immune response elicited after a *Leptospira* pre-infection persists over time and leads to protection against subsequent infection depending on species. The present study of the long-term evolution of humoral response demonstrates that C57BL6/J mice mount a sustained and effective long-lasting immune response after experimental infection with different strains of virulent leptospires or immunization with attenuated strains or bacterin.

We monitored the antibody evolution over a six-month period after infection which could be considered as a good assessment of the long-term humoral response in mice. Because gender differences in susceptibility to leptospirosis have been shown in sensitive hosts like hamsters [61] and humans [2], with males presenting more severe symptoms, we studied both females and males, but we did not find in the mouse model any differences in severity of the disease nor Ig responses. However, the differences in disease severity previously observed in male hamsters did not translate into a differential humoral response, since no significant difference in IgG responses was measured between male and female hamsters [61]. Likewise, for both Manilae and Icterohaemorrhagiae serovars upon the first infection, there was no effect of the inoculum dose on the IgG production, which suggests that the magnitude of the IgG response is not correlated to the severity of the disease, in contrast to their primary IgM or IgA productions that were linked to the dose, when they occurred.

Here we studied three different strains/serovars responsible for severe human leptospirosis worldwide. In our previous study in which we investigated innate responses in mice, we did not observe major differences between these three strains [62]; therefore, the differential response observed in the antibody production after the primary infection was unexpected. The first striking difference was the different Ig profiles obtained with the virulent Icterohaemorrhagiae Verdun Cl3 strain compared with the two other serovars. The magnitude of the Ig responses was very low compared to the Manilae and Copenhageni serovars. Because the humoral response is believed to be directed mostly at the O antigen part of the LPS, we wondered whether the Icterohaemorrhagiae Verdun strain would have a shorter LPS like the M895 Manilae strain that also did not elicit strong responses [47]. However our recent study showed

that the Icterohaemorrhagiae Verdun LPS was similar in size to those of the Copenhageni and Manilae strains [63]. Since the transient IgM and low IgG responses induced by the Verdun Cl3 strain mimicked either the heat-inactivated Manilae L495 strain, the L495 at low dose or the Patoc strain, and since mice showed no clinical signs upon Icterohaemorrhagiae infection, we also surmised that the Verdun Cl3 might have lost its virulence. However, we showed that the Verdun Cl3 strain had not lost its pathogenicity and produced serious disease when injected at higher dose in mice. Surprisingly, even at the high dose, the IgM and IgG response remained low and identical to that seen with the $2\times10^7$ dose. In addition, the results about cross-reactivities showing that other serovars can recognize Verdun antigens strongly suggest the specific capacity of Verdun virulent strain to dampen the specific humoral response. Altogether, these results suggest that the mouse immune system reacts differently to infection with the Icterohaemorrhagiae Verdun compared to the two other serovars. The basis for this difference remains unknown.

Second, we identified a sustained IgM response that occured for the virulent Manilae L495 and Copenhageni Fiocruz LV2756 strains, but not for the Icterohaemorraghiae Verdun Cl3 strain. Interestingly, we showed that the Verdun Cl3 strain did not colonize the mouse kidney, although both Manilae and Copenhageni did so, confirming previous data [20,54,64,65]. This lack of renal colonization by the Verdun Cl3 was not expected considering the clinical features of severe leptospirosis recorded after high dose infection and determined using an evaluation scoring, including weight loss and clinical signs. This is surprising, since other murine rodents such as rats are known worldwide as renal carriers of a large array of *Leptospira* serovars, including Icterohaemorrhagiae [15]. However, although the Icterohaemorrhagiae Verdun strain is pathogenic [66,67], and like other serovars may cause pulmonary hemorrhages, a syndrome also frequently described in guinea pigs, and reported associated with the most severe human leptospirosis cases [68], it has also been reported that experimental infection with Icterohaemorrhagiae strains in outbred mice and hamsters results in subclinical infections [67]. Host-species differences of *Leptospira* innate sensing have already been described between mice and humans [69–71], but potential differences between rats and mice have not yet been investigated. It would be interesting to use a more mouse-adapted serovar such as Arborea or Australis [72] and repeat these experiments to test whether these leptospires colonize the mouse kidneys and also elicit sustained IgM. Since the Manilae M895 attenuated mutant, like the Icterohaemorrhagiae Verdun strain, elicited a classical transient IgM response peaking at D15 and was not found in kidneys, and because we found in one mouse a discrete renal colonization and residual IgM response one month post-infection with the Icterohaemorrhagiae Verdun Cl3 at high dose, and Manilae L495 at low dose, we hypothesize that the renal colonization was the cause of sustained IgM response. This also confirmed our previous results obtained with the Manilae L495 derivative bioluminescent strain, showing that an infectious dose of $10^6$ bacteria/mouse always led to renal colonization, but a low dose of $10^5$ was less effective [54]. In addition, using live imaging of a bioluminescent derivative of the Patoc 1, we also showed the lack of persistence of this strain in mice [70], that potentially may correlate with very low and transient levels of IgM.

How could renal colonization explain the atypical sustained IgM production? We could hypothesize either that the leptospires constantly re-infect the host via urinary shedding, or that leptospires could re-enter the circulation from the proximal tubule. In both situations, leptospires would be opsonized with specific antibodies leading to their immediate clearance, and as in a new infection, would trigger an IgM response. Chronicity of infections has been associated with sustained levels of IgM, for example in *Toxoplasma gondii* infections. In France, pregnant women are systematically tested for this parasite in the first trimester of pregnancy in order to detect infections that are highly dangerous for the fetus. Quite unexpectedly, around

5% of women tested harbor high and sustained levels of IgM (invalidating the diagnosis of a recent infection) supposedly due to frequent reinfections because of consumption of contaminated food, but it is also possible that it may due to asymptomatic toxoplasmosis leading upon immune depression during gestation to the reactivation of cysts in the brain and release of tachyzoites [73,74]. In the case of leptospirosis, sustained IgM responses have been recorded in dogs [75] that are known to excrete *Leptospira* in their urine [76]. On the other hand, only transient IgM responses have been measured in bovines experimentally infected with *L. interrogans* serovar Hardjo [77]. If our hypothesis is correct, it suggests that the route/ infectious dose/ or serovar ($10^9$ bacteria, intramuscular) was not appropriate to provoke renal colonization. In fact in bovines, even though *Leptospira* shedding in urine is frequently reported [78], leptospirosis may also present as a genital disease, distinct from the chronic renal disease observed in other animals. However, we can speculate that what is often considered false IgM positive results in apparent healthy animals could be due to asymptomatic renal colonization [79]. Our results could also potentially explain the poor accuracy of IgM testing to detect acute leptospirosis, evidenced in human populations in endemic areas [79]. Although humans are not usually long term renal carriers, a persistent IgM response has been measured several months or years after the onset of leptospirosis in humans [30,80–82]. Whether this could reflect a subclinical and asymptomatic renal carrier status [83] as observed in endemic area of leptospirosis [84,85] remains to be studied.

We also highlighted another difference of sensing between serovars, as a strong and persistent IgA response was observed at the $2\times10^7$ dose with the virulent Copenhageni Fiocruz LV2756 strain, but not with the Manilae L495 and the Icterohaemorrhagiae Verdun strains, although an equivalent transient IgA response was observed with higher doses of the virulent strains of all serovars used to challenge the PBS controls. This indicates that compared to other serovars, the IgA response is triggered by the Copenhageni Fiocruz LV2756. This clinical strain was isolated from a patient lung [48]. We did not study the pathophysiology of the Copenhageni Fiocruz strain LV2756 in the mouse lung, but we already showed that L1-130, another Copenhengani Fiocruz strain used worldwide in experimental infections, disseminates in lungs during the acute phase of the infection [28]. Leptospirosis due to the Copenhageni serovar often presents in humans as Severe Pulmonary Hemorrhagic Syndrome (SPHS) [86], which has been associated with deposit of Ig, in particular IgM, IgG and IgA, in humans [87]. Whether the IgA secretion would participate in the pathophysiology, or in contrast, in the resolution of the infection is not clear and warrants further studies. Since leptospires naturally infect through skin or mucosa, it should not be surprising to detect IgA, considering the predominant role of IgA in the host mucosal defense. Accordingly, serum IgA has been found to persist several months in Brazilian patients diagnosed with leptospirosis [81,88,89]. However, in our case, we injected the bacteria intraperitoneally, bypassing the skin or mucosal barriers, which potentially could explain the lack of IgA response to the Manilae and Icterohaemorrhagiae infections, but does not explain the results obtained with the Copenhageni strain. Alternatively, this may suggest that the IgA response could be specific to the Fiocruz pathophysiology. Interestingly we recently showed that live leptospires (Manilae L495, Copenhageni Fiocruz L1-130 and Icterohaemorrhagiae Verdun Cl3) escape the TLR5 recognition *in vivo* in mice and *in vitro* in human cells [71]. However, once heat-killed or degraded by antimicrobial peptides, only the Copenhageni strain was able to signal through mouse-TLR5 [71]. Mouse-TLR5 activation has been associated with anti-flagellin IgA secretion [90]. Whether the species-specificity of the IgA response observed here relies on TLR5 recognition requires further studies. Other routes of experimental infection, such as the transdermal route [52], should be tested with these different serovars to check whether it influences the IgA humoral response.

Then, we studied the recall humoral response one or two months post-challenge performed six months post-infection or immunization with attenuated strains or bacterin. Although IgA was found at low but sustained levels only in mice primarily infected with the virulent Fiocruz strain, higher levels of IgA were present in all mice after challenge, demonstrating a memory independent of renal colonization.

The subclasses of IgG have different characteristics and effector functions, through binding to the complement and/or to Fc-gamma receptors on phagocytes [91]. Therefore, the differential profiles obtained here may be informative. IgG1 and IgG3 are usually directed against proteins and depend on the T cell / B cell MHC-class II presentation, whereas polysaccharide antigens such as LPS induce IgM [25,27–30] and class switching to IgG2 [91]. Consistent with the presence of LPS in leptospires, we observed an overall IgG2 response in primary infected mice that was notably boosted after challenge, including for Verdun that did not trigger much IgG at the primary infection. In addition, the Manilae M895 mutant harboring a truncated LPS elicited high IgG responses. These results were expected, since the LPS masks the outer membrane proteins and lipoproteins that potentially can elicit IgG. However, other subclasses of IgG were also elicited with the virulent strains, and also to a lesser extent with the attenuated mutant, suggesting that in addition to carbohydrate antigens from the LPS, protein antigens were also recognized. All IgG subclasses were notably boosted after challenge. Interestingly, compared to the virulent Manilae L495 strain, the M895 LPS mutant stimulated a higher or equivalent production of IgG1, and an equivalent peak of IgG3, although the latter did not persist after challenge. An interesting point is that in mice infected with the Manilae M895 mutant, IgA and IgG3 were considerably boosted post-challenge. This could explain the protective and vaccine effects of this M895 attenuated strain (and potentially another LPS mutant) in the hamster model [44,47]. The low doses of L495 also triggered p.C. a boost in IgG3, whereas heat-killed bacteria triggered a boost in IgG2c, which is not surprising considering the fact that heating at 100°C preserved the LPS antigens but may have altered the protein antigens. Interestingly, the boost in IgG3 p.C. was higher for the Manilae strains compared to Fiocruz. We may speculate that the IgG3 response could be important to clear the leptospires from the kidneys.

Cross-reactivity experiments showed that infection with the Fiocruz strain triggered cross-reactive antibodies against both Manilae L495 and Icterohaemorrhagiae Verdun strains. This is in line with the recent study showing that the attenuated live Fiocruz *fcpA* mutant confers cross-protection against infection with other serovars, including Manilae [49]. Here, we show that the Fiocruz strain triggers high amounts of IgG1 and IgG3, which by large recognize proteins. However, if these IgG1 and IgG3 cross-reacted with the Icterohaemorrhagiae serovar, they do not cross-react with Manilae, although in contrast, IgG2 elicited with Fiocruz cross-reacted with both Manilae and Verdun. These results are difficult to reconcile with the data showing that the antigenic cross-protection of *fcpA* was due to antibodies directed against highly conserved proteins of leptospires, rather than neutralising antibodies directed at serovar specific LPS [49].

One or two months after challenge, we measured the leptospiral loads in kidneys of control and pre-infected mice. As expected, the M895 mutant that has shown potential as a live attenuated vaccine, protected against kidney colonization upon challenge. The interpretation of results obtained with the virulent Manilae L495 and Copenhageni Fiocruz LV2756 pre-infected mice is more complicated. Indeed for Manilae L495, we did not find leptospires in the kidneys. At the $2\times10^7$ dose it was surprising since although it was in a different experiment, we had renal colonization one month p.i., and we previously showed that in these conditions the renal colonization is stable and lifelong [54]. Therefore it would suggest that the boost in the humoral Ig might be potent enough to clear the leptospires from the kidneys. It remains to be

seen whether these elevated IgM, IgA and IgG antibodies could clear and prevent reinfection from the renal tubules. This would be very interesting as a novel therapeutic strategy. However, conversely, we were as surprised by the higher levels of leptospires found in kidneys of mice pre-infected with Copenhageni, compared to the control mice. This would suggest that the leptospires from the primary infection accumulated with the leptospires coming from the challenge, which suggests that the Copenhageni defense may not be as efficient as the one elicited by Manilae L495. However, we did not observe any major differences in the Ig responses, after the primary infection or after the challenge that could explain such a difference. Further studies are required to understand this striking observation. Whether it could be linked to differential post-translational modifications already observed in the proteome of leptospires in rat kidney [92] that may help the Copenhageni Fiocruz LV2756 but not the Manilae L495 to escape the humoral response warrants further comparative investigation.

We demonstrated that immunization with virulent, attenuated or heat-killed leptospires all induced a long-term immunity that was protective and efficient in alleviating severe disease in mice after a subsequent homologous infection. This is in line with the fact that bacterins are efficient serovar-specific vaccines [24] and that passive transfer of serum of infected mice protected naive mice from severe leptospirosis [20,28]. Similarly, administration of human IgM [30] or a monoclonal antibody directed against the LPS [26] protected guinea pigs. However, injection of Manilae heat-killed bacterin triggered only a weak Ig response. It could suggest that immunity to leptospirosis may not be limited to the humoral response and protection could also be cell-mediated as demonstrated in bovines after vaccination [93]. Indeed, peripheral blood mononuclear cells (PBMCs) from cattle immunized with inactivated leptospires of serovar Hardjo proliferated and produced IFN-γ mainly from CD4+ cells after *in vitro* stimulation with different leptospiral antigens [93–95]. Production of cellular immunity Th1 effectors as IFN-γ, IL-12 and TNF-α was also observed in human whole blood after *in vitro* stimulation with heat-killed leptospires [96,97]. This cellular immunity could explain why no correlation was observed between the 20% seroprevalence determined in sampled cattle and the leptospires excretion in urine measured in only 7% of them [98]. On the other hand, it was shown that even subagglutinating levels of antibodies could opsonise leptospires for clearance [26], suggesting that this response *per se* might be protective. Studies of cellular immunity in leptospirosis remain to be addressed in both hamsters and mice.

Our data suggest that the mounting of the adaptive antibody response was effective for all strains, even though it was weaker upon Icterohaemorrhagiae Verdun infection. Recent work from our laboratory suggests that leptospires, including Verdun, trigger an incomplete TLR4 response in mice. Whether this partial escape of leptospires from mouse TLR4, or its total escape as shown in human cells [55] is involved in this low humoral response is an interesting question that is not easy to address in mice. Like humans, TLR4 knock-out mice are sensitive and can die from acute leptospirosis, and we showed that TLR4 has a crucial role in mounting the protective anti-Copenhageni IgM response [28]. Other studies in C3H/HeJ mice naturally not expressing TLR4 have described humoral and cellular responses upon experimental leptospirosis with Copenhageni Fiocruz L1-130 [17], but only total, not *Leptospira*-specific Igs have been studied [52,65,99]. Considering the predominant role of TLR4 in immunity, the caveat of these studies is that TLR4 deficient mice are immune deficient, which interferes with the interpretation of results. However, a recent study using humanized TLR4 mice [100], which are immune competent mice, showed that those mice are not more sensitive to acute leptospirosis, but rather suggested that a functional TLR4 receptor being of mouse or human was required to protect the mice against leptospirosis. Whether the immunoglobulin response would have been different has not been investigated since only the first 15 days post-infection have been studied [101].

In conclusion, as for many infections, analyses of antibody responses in *Leptospira* infection are of prime importance for diagnosis, but also for vaccine development, as well as to provide some insights into pathogenicity mechanisms characterizing the microorganism [24]. In the *Leptospira* field, correlates of protection and basic knowledge about the humoral responses after *Leptospira* infection are lacking. Although our study concerns mice that are resistant animal to leptospirosis, and thus different from humans who are susceptible to develop an acute life-threatening disease, the mouse model will be useful to address the mechanisms linked to long-term protection and potentially understand why there is not such a long lasting protection in human leptospirosis. In addition, although we were not able to associate a specific profile with protection, this study revealed important findings, such as the link between sustained IgM and renal colonization, and the potential role of IgG3 in renal clearance that needs further investigation. In conclusion, our study brought a better understanding of host immunity against *Leptospira* and new knowledge about the humoral responses toward leptospires in mice that depend on serovar and physiopathology.

## Supporting information

**S1 Fig. Outcome evaluation of leptospirosis and specific anti-*Leptospira* Ig isotype according to gender.** Outcome evaluation after pre-infection or subsequent infection in C57BL/6 mice with virulent strains or avirulent mutants representative of 3 distinct pathogenic (Manilae, Copenhageni, Icterohaemorrhagiae) serovars. A) Weight evolution individually recorded (n = 5 mice per group) in mice inoculated with $2\times10^7$ of virulent Manilae L495 strain or M895 mutant, or virulent Copenhageni Fiocruz LV2756 strain, or virulent Icterohaemorrhagiae Verdun (2 doses) or avirulent (AV) strains, or PBS as negative control. Weight changes from the initial weight at the day of infection, expressed as a percentage, was daily recorded from D0 to D7 p.i., then weekly (D8 to the end of the experiment). Graphs represent the mean ± SD of the weight change recorded overtime for each experimental group. B) Colored score values individually recorded at defined time-points after infection (Days p.i.) as in A) and after homologous challenge (Days p.C.). n = 5 mice /group, identified M1 to M5 in each group. C) List of clinical signs recorded after infection with *Leptospira* to establish individual severity scores as indicated in the color code. A score value equal to or greater than 7 out of 11 defines an end point limit. D) Profiles of specific Ig (left panel), IgM (central panel) and IgG (right panel) for 1 month in male or female mice (n = 5/group), after experimental infection with $2\times10^7$ virulent strains of Manilae, Copenhageni and Icterohaemorrhagiae serovars, or PBS as negative control.
(TIF)

**S2 Fig. Pre-infection with Fiocruz LV2756 and Manilae L495 virulent strains leads to cross-reactive IgG subclasses, unlike Icterohaemorrhagiae Verdun.** Profiles of different IgG subclasses of anti-*Leptospira* immunoglobulins tested against antigens from heterologous serovars. Specific IgG1 (first row), IgG3 (second row), IgG2b (third row) and IgG2c (bottom/fourth row) immunoglobulins obtained after experimental infection with $2\times10^7$ virulent leptospires representative of 3 distinct serovars, Manilae L495 (left column), or Copenhageni Fiocruz LV2756 (central column) or Icterohaemorrhagiae Verdun (right column), were checked against heterologous serovar antigen preparation (dashed line) and compared to the profile obtained with homologous serovar (full line). Anti-*Leptospira* IgG subclasses were determined by ELISA assay using specific *Leptospira* antigen preparations and appropriate dilutions of serum collected at determined time-point after infection. Each figure represents the profiles for specific IgG1, IgG3, IgG2b and IgG2c subclasses obtained from serum tested in pool (for a same experimental group). Antibody responses were assessed up to day 180 p.i. with female

mice (n = 5).
(TIF)

**S3 Fig. Bacterins or low dose pre-infection induce a moderate but efficient anti-*Leptospira* humoral memory response.** Outcome monitoring and specific Ig responses in C57BL/6J mice intraperitoneally pre-infected with different doses of avirulent or inactivated Manilae L495 and, after 6 months, challenged with the Manilae bioluminescent derivative MFLum1 strain. A) Relative light units (RLU) measured by live imaging (IVIS) in mice pre-infected with $2\times10^7$ or $1\times10^5$ of virulent L495 strain or $2\times10^7$ of M895 mutant or $2\times10^7$ of heat-killed L495 or PBS and challenged with $5\times10^8$ of MFLum1 strain (left panel). The IVIS has been performed as recently described [102], at 30 min, D3, D15, D28 and D55 post-challenge (p.C.). The graph represents the mean ± SEM of the average radiance in n = 5 mice in each group, imaged in ventral (D0 and D3 p.C.) then in dorsal position, and gated on the whole body. The background level of light was measured on control C57BL/6 mice injected with PBS at the time of infection (PBS). Representative images of $2\times10^7$ Manilae L495 or PBS pre-infected mice challenged with MFLum1 and then, tracked by IVIS at 30 min, D3 and D15 post-challenge (p.C.) (right panel) with n = 5 mice /group. The blue to red scale is proportional to the intensity of bioluminescence, reflecting the number of live leptospires. B) Profiles of specific anti-*Leptospira* isotype (IgM, IgA) and C) IgG subclass responses (IgG1, IgG3, IgG2b, IgG2c) produced in mice inoculated with $2\times10^7$ (blue line) or $1\times10^5$ (dashed blue line) of alive virulent L495 strain or $2\times10^7$ (grey line) of heat-killed L495 or PBS as control, then challenged with $5\times10^8$ MFLum1. Specific Ig responses were determined by ELISA immunoassay using specific *Leptospira* antigen preparation and appropriate dilutions of serum collected at defined time-point after challenge. Each figure represents the profiles for each type of Ig obtained from serum tested in pool (for a same experimental group with n = 5 mice /group). Antibody responses were assessed up to day 60 p.i.
(TIF)

## Acknowledgments

We thank Elsio Wunder and Albert Ko for providing the clinical Fiocruz LV2756 strain. We thank Brigitte David-Watine for critical reading of the manuscript.

## Author Contributions

**Conceptualization:** Frédérique Vernel-Pauillac, Catherine Werts.

**Formal analysis:** Frédérique Vernel-Pauillac, Catherine Werts.

**Funding acquisition:** Ivo G. Boneca.

**Investigation:** Frédérique Vernel-Pauillac, Catherine Werts.

**Methodology:** Frédérique Vernel-Pauillac.

**Project administration:** Catherine Werts.

**Resources:** Gerald L. Murray, Ben Adler.

**Supervision:** Catherine Werts.

**Validation:** Frédérique Vernel-Pauillac, Catherine Werts.

**Visualization:** Frédérique Vernel-Pauillac.

**Writing – original draft:** Frédérique Vernel-Pauillac, Catherine Werts.

**Writing – review & editing:** Frédérique Vernel-Pauillac, Gerald L. Murray, Ben Adler, Ivo G. Boneca, Catherine Werts.

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
