## [Decision Letter · Decision Letter 0]

12 Jan 2021

Dear Dr. Werts,

Thank you very much for submitting your manuscript "Anti-Leptospira immunoglobulin profiling in mice reveals strain specific IgG and persistent IgM responses associated with virulence and renal colonization" for consideration at PLOS Neglected Tropical Diseases. As with all papers reviewed by the journal, your manuscript was reviewed by members of the editorial board and by several independent reviewers. The reviewers appreciated the attention to an important topic. Based on the reviews, we are likely to accept this manuscript for publication, providing that you modify the manuscript according to the review recommendations. 

Sincerely,

Brian Stevenson, Ph.D.

Associate Editor

Richard Phillips

Deputy Editor

Reviewer's Responses to Questions

**Key Review Criteria Required for Acceptance?**

**Methods**

-Are the objectives of the study clearly articulated with a clear testable hypothesis stated?

-Is the study design appropriate to address the stated objectives?

-Is the population clearly described and appropriate for the hypothesis being tested?

-Is the sample size sufficient to ensure adequate power to address the hypothesis being tested?

-Were correct statistical analysis used to support conclusions?

-Are there concerns about ethical or regulatory requirements being met?

Reviewer #1: The methods are fully described and are appropriate for addressing the overall study objective.

Reviewer #2: Study design, sample size, methods employed and statistical analyses are adequate.

Reviewer #3: -The objective of the study was clearly stated: To assess the long-term antibody response (in mice) and protection against subsequent infection with various Leptopspira strains.

-Yes, the appropriate concentration of bacteria was used for infection experiments and standardized laboratory methodologies were used to measure bacterial loads and antibodies, RT-PCR and ELISA, respectively.

-The mouse model is the appropriate system since infection of mice with Leptospira is not fatal, allowing long-term measurement of the antibody response. 

-The sample size and statistics used to measure differences could use some clarification in the methodology.

-No ethical or regulatory concerns.

**Results**

-Does the analysis presented match the analysis plan?

-Are the results clearly and completely presented?

-Are the figures (Tables, Images) of sufficient quality for clarity?

Reviewer #1: Overall the results are clear and well presented. Comments on the results section are relatively minor:

 1. Lines 336-338 and discussion: to understand the significance of the differing response to the infectious dose, the authors should indicate, if known, the expected dose during a natural infection.

2. Figure 3B, left panel: why did two previously infected mice have a higher leptospiral organ burden? Was there a different immune response detected in these two outlier mice?

Reviewer #2: Results, figures and tables are clearly presented.

Reviewer #3: -Yes

-Yes

-Yes

**Conclusions**

-Are the conclusions supported by the data presented?

-Are the limitations of analysis clearly described?

-Do the authors discuss how these data can be helpful to advance our understanding of the topic under study?

-Is public health relevance addressed?

Reviewer #1: There appears to be a disconnect between the findings shown in Figure 1A and Figure 3A, in that the avirulent serovars provided a minimal humoral response but still protected the mice from subsequent homologous infection. On Line 647-650 the authors offer an explanation for a similar result obtained with heat-killed Manilae; do the authors expect cell-mediated immunity also plays a role in the observed immunity with the avirulent serovars?

Reviewer #2: Conclusions are supported by the data presented. Discussion covers comparison and how this study is important.

Reviewer #3: -The conclusions are supported by the data.

-The limitations of the study could be better communicated.

-Yes.

-Relevance to public health was addressed but not directly.

**Editorial and Data Presentation Modifications?**

Reviewer #1: 1. Lines 191 and 421: to conform with standard nomenclature for animal experimentation, it is recommended that the word “killed” be replaced with “euthanized”.

2. Figure 3A: it is recommended that the Y axis be expanded to decrease the clustering of the data points, and to increase the ability to view these data points, especially at the early time periods.

Reviewer #2: (No Response)

Reviewer #3: I did not catch any errors, I would say the authors are more than capable of polishing the text as needed.

**Summary and General Comments**

Reviewer #1: The manuscript “Anti-Leptospira immunoglobulin profiling in mice reveals strain specific IgG and persistent IgM responses associated with virulence and renal colonization” by Vernel-Pauillac et. al. reports on the differing immunological response of mice infected with 3 different serovars of Leptospira to advance the understanding of the correlates of protection against disease. Overall this manuscript provides an important contribution to the Leptospira field of study and provides insight relevant to vaccine development.

Reviewer #2: This manuscript by Vernel-Pauillac et al provides some interesting information about immune responses in C57BL/6J mice to different strains leptospires belonging to three serovars. Different immunoglobulin types/subtypes were measured over a period of time. Mice were challenged with homologous bacteria. This study is important in understanding immune responses to leptospiral infection and may have implications on improving diagnostics. Here are some questions/comments:

1. Since the main focus of the study is measuring immunoglobulin types/subtypes, more information is needed on the ELISA used in this study, eg., (i) was this assay developed/validated in-house? Please provide a reference(s) in either case. (ii) is there any known cross-reactivity among different anti-mouse antibodies used in this study? Please provide a reference if available.

2. Is it possible that the observed relationship between persistent peripheral IgM levels and renal colonization has something to do with the immune environment of kidneys and this phenomenon has some similarity to immune deviation seen during infection of sites like brain, eyes, testes etc.?

PLOS authors have the option to publish the peer review history of their article (what does this mean?). If published, this will include your full peer review and any attached files.

Reviewer #1: No

Reviewer #2: No

Reviewer #3: Yes: Azad Eshghi
---

## [Editor Report · Decision Letter 1]

23 Feb 2021

Dear Dr. Werts,

We are pleased to inform you that your manuscript 'Anti-Leptospira immunoglobulin profiling in mice reveals strain specific IgG and persistent IgM responses associated with virulence and renal colonization' has been provisionally accepted for publication in PLOS Neglected Tropical Diseases.

Best regards,

Brian Stevenson, Ph.D.

Associate Editor

Richard Phillips

Deputy Editor

---

## [Editor Report · Acceptance letter]

8 Mar 2021

Dear Dr. Werts,

We are delighted to inform you that your manuscript, "Anti-Leptospira immunoglobulin profiling in mice reveals strain specific IgG and persistent IgM responses associated with virulence and renal colonization," has been formally accepted for publication in PLOS Neglected Tropical Diseases.

Best regards,

Shaden Kamhawi

co-Editor-in-Chief

Paul Brindley

co-Editor-in-Chief
